# OpenReview forum: "Training Language Models to Win Debates with Self-Play Improves Judge Accuracy"
_ICLR.cc/2025/Conference — Submitted to ICLR 2025_

### Official Review · Reviewer_3vo5 · 2024-10-22

**Soundness:** 2
**Presentation:** 1
**Contribution:** 2
**Rating:** 3
**Confidence:** 2

**Summary:**

This work explores the use of debate as a scalable oversight method for language models. The proposed method trains models to debate using self-play and assess whether this approach enhances the accuracy of AI-based evaluators in judging complex tasks, such as reading comprehension. The key contribution of the paper is demonstrating that models trained to debate in adversarial settings lead to more accurate judgments by AI evaluators compared to non-adversarial consultancy models.

**Strengths:**

1. This paper introduces debate as a method for scalable oversight by leveraging self-play for training language models, which shows promise for improving evaluator accuracy in complex tasks.

2. This paper discusses and compares multiple baselines (e.g., single, ensembled, and double consultancy).

**Weaknesses:**

1. The paper's experiments are limited to reading comprehension tasks, which raises concerns about the generalizability of the findings to other complex reasoning domains. Could the effectiveness of debate training extend to tasks beyond reading comprehension?

2. The debate protocol is limited to a **two-turn** setting, which restricts the depth of argumentation and refutation between debaters. This simplified structure may not fully capture the complexities of real-world debates, where extended back-and-forth exchanges are often necessary to expose subtle flaws in reasoning. As a result, the findings may overestimate the effectiveness of debate in fostering accurate judgments, since more intricate discussions could reveal different dynamics or weaknesses in the model's performance​.

3. While the experiments provide some insights, they are not entirely convincing. The study only employs GPT-4-Turbo as the judge and Llama3-8B-Instruct as the debate and consultancy models. Evaluating the proposed method across a broader range of model architectures and sizes would strengthen the assessment of its generalizability.

4. Although debate is presented as a mechanism for scalable oversight, the paper finds little evidence that explicit refutation materially affects the judge’s decision-making. This undermines one of the key proposed benefits of debate as an oversight mechanism​.

**Questions:**

1. See weaknessnes.

2. Figures 1, 2, and 5 are not referenced in the text. It would be helpful to cite them in the relevant sections to enhance clarity and support the descriptions.

---

### Official Review · Reviewer_dtKg · 2024-10-27

**Soundness:** 3
**Presentation:** 3
**Contribution:** 3
**Rating:** 6
**Confidence:** 4

**Summary:**

The paper investigates the robustness of training debater models to enhance the accuracy of judgments in long-context reading comprehension tasks. It introduces a practical and emerging scenario called "super-alignment" or "scalable oversight" where it is assumed that AI systems may eventually surpass the wisdom of human supervisors  and underscores the need to explore methods by which a weaker model could effectively supervise a stronger one. To emulate this, the paper creates an asymmetric information environment, allowing the debater model access to the full context while withholding it from the judgment model. The findings suggest that debate training fosters the development of stronger, more informative arguments, indicating its potential for delivering high-quality supervision in tasks that are challenging to evaluate directly.

**Strengths:**

1. The paper introduces a novel experimental design to simulate information asymmetry, rigorously testing their approach by comparing the impact of the debate process with a consultancy process on the final judgment accuracy.

2. The findings are intriguing, revealing a positive relationship between a model’s ability to win debates and the usefulness of its debate transcripts for uncovering correct answers. In contrast, the non-adversarial "consultancy" approach shows no positive correlation between the judge's accuracy and the skill level of the consultant.

3. These findings have the potential to shape future research on "super alignment" or "scalable oversight" by establishing a foundational debate-then-judge framework.

**Weaknesses:**

While I find the experimental setting intriguing, some aspects of the evaluation are unclear:

1. In Section 2.4, the use of "self-play" to evaluate judge accuracy is ambiguous. From my perspective, there shouldn’t be a need for the judge to "self-play." Instead, judge accuracy should be evaluated by its ability to correctly discern outputs from the debater and consultant. Specifically, if the judge evaluates a debater advocating for an incorrect answer as incorrect, it should be marked as correct, and conversely, if it rates a debater with a wrong answer as correct, it should be marked as incorrect, and vice versa.

2. In line 187, the method for calculating the consultant win rate remains unclear. If the consultant is assigned to defend an incorrect answer, under what conditions is this considered a "win"? Similarly, if the consultant defends a correct answer, what conditions would lead to a "loss"?

3. There is insufficient detail on the training of the judges, including the instances in the training dataset and the format of input and output. Is there evidence to support line 215, which states that judge decisions “were no longer clustered at the boundaries as they were prior to fine-tuning”?

4. In line 473, the conclusion that "explicit refutation does not yet seem to play a role in judge decision-making in our setting" is not sufficiently substantiated and lacks experimental evidence to support it.

**Questions:**

1. In Figure 3 (right), what does the judge Brier score represent?

2. In line 264, how is a "win" or "loss" response determined when constructing the DPO training datasets?

3. In line 313, what does "confidence" refer to in the context of the reward function? Was GPT-4 Turbo trained to output a confidence level?

4. In line 318, what do the two win rates in parentheses represent?

5. In line 362, why are there three pairs per round and two rounds per question?

---

### Official Review · Reviewer_VXq5 · 2024-11-04

**Soundness:** 3
**Presentation:** 3
**Contribution:** 3
**Rating:** 6
**Confidence:** 3

**Summary:**

This paper studies that training language models to win debate can enhance the accuracy of evaluator judgments. Specifically, the authors use the QuALITY dataset as a testbed, where judges assess debate transcripts to determine whether the debater’s answer is correct. They train a GPT4-based judge model and develop a Debate-Powered Optimization (DPO) method for training the debate model with LLAMA3. Results demonstrate that debate training enhances the model's ability to produce stronger, more persuasive arguments, which in turn improves the accuracy of the judge model. In contrast, non-adversarial methods do not exhibit this positive correlation.

**Strengths:**

- The overall idea of exploring debate model training to enhance evaluator accuracy is both novel and interesting;
- The paper offers comprehensive analyses demonstrating the correlation between debate model and judge model performance across both debate and non-adversarial settings, potentially providing insights for future research on multi-agent debate systems.

**Weaknesses:**

- Limiting experiments to a single dataset and task (QuALITY) may affect the generalizability of the conclusions. Incorporating more different tasks could provide broader insights for the community.
- The paper lacks evaluations on fine-grained aspects, such as factuality. For instance, it would be valuable to assess whether debate training results in more factual transcripts and whether the judge model improves its ability to detect factual discrepancies.
- The judge model is based on GPT-4. How would results differ if a smaller language model, such as LLAMA3-8B, were used as the judge model?

**Questions:**

- In the round-robin tournament for the debate setting: are the models on both sides identical checkpoints from the debate-trained model, or is one side represented by the debate-trained model while the other serves as a baseline (e.g., without debate training)?

---

### Official Review · Reviewer_G6gV · 2024-11-04

**Soundness:** 3
**Presentation:** 1
**Contribution:** 2
**Rating:** 3
**Confidence:** 3

**Summary:**

In an LLM as judge scenario, where LLMs act as judge, by evaluating a task output by looking at other agents debating or consulting, this paper shows that agents that are trained using a modified DPO method help the judge in more accurately evaluating the task. They show this using a modified GPT-4 as a judge, and debate/consultancy trained LLAMA 8B models are either debaters or consultants. They also show that training consultants doesn’t improve the judge's accuracy as a trained debater while closing the gap for untrained debater.

**Strengths:**

This work builds upon the previous work such as Michael at al and Kenton et al to show that trained debaters outperform trained consultants.

The authors introduce a novel DPO based training algorithm to train models to act as better debaters.

The debaters and corresponding methods are evaluated on a hard subset of the QuALITY dataset (Pang et al., 2022) following Parrish et al. (2022b).

**Weaknesses:**

The writing - specifically in the abstract and introduction are very dense and hard to read.

Most of the findings in this paper seemed similar to the findings from Kenton et al (2024) work and the writing doesn’t make a clear distinction how this work is different either in the main paper or the related work section.

The evaluation was also done only on QuALITY dataset, and that also made it hard for me to compare this work w.r.t to the contributions presented in Kenton et al (2024), which uses a broader set of evals to evaluate this paper in a more comprehensive manner.

**Questions:**

Covered above

---

### Comment · Area_Chair_pj82 · 2024-11-27

Dear reviewers,

Thank you for your efforts reviewing this paper. If you haven't, can you please check the authors' responses and see if your concerns have been addressed? Please acknowledge you have read their responses. Thank you!

---

### Meta-Review · Area_Chair_pj82 · 2024-12-19

**Metareview:**

Summary:

This paper studies the robustness of debate as a method of scalable oversight, by training models to debate with data generated via self-play. It finds that language model based evaluators answer questions more accurately when judging models optimized to win debates in a long-context reading comprehension task. The paper finds no such relationship for consultancy models trained to persuade a judge without an opposing debater present. It conducts quantitative and qualitative comparisons between the debate models and novel consultancy baselines, and finds evidence that debate training encourages stronger and more informative arguments, which can help provide high-quality supervision for tasks that are difficult to directly evaluate.

Strengths:

Reviewers generally agree that the ideas introduced in the paper to explore debate model training to enhance evaluator accuracy are novel. The findings are intriguing and have the potential to shape future research.

Weaknesses:

The major weaknesses reviewers find in general can be summarized into two aspects:

1. Writing. For example, the abstract and introduction are hard to read, and the differentiation from existing work is not clear.

2. Evaluation: (1) the focus on only one dataset, (2) limitations to the LLama3-8B-Instruct model. These limitations make the results (despite interesting) not completely convincing.

**Additional Comments On Reviewer Discussion:**

Many weaknesses pointed out by reviewers (e.g., those by Reviewer dtKg ) are due to unclarity in the writing, and the authors have clarified them during the rebuttal period. However, the paper does need to further improve the paper writing and address the limitations in the experiments and evaluation. I believe it will benefit from another round of revision.

---

### Decision · Program_Chairs · 2025-01-22

Reject